# Migratory Tumor Cells Cooperate with Cancer Associated Fibroblasts in Hormone Receptor-Positive and HER2-Negative Breast Cancer

**DOI:** 10.3390/ijms25115876

**Published:** 2024-05-28

**Authors:** Eun Hye Joo, Sangmin Kim, Donghyun Park, Taeseob Lee, Woong-Yang Park, Kyung Yeon Han, Jeong Eon Lee

**Affiliations:** 1Samsung Genome Institute, Samsung Medical Center, Seoul 06351, Republic of Korea; thegracejoo@gmail.com (E.H.J.); woongyang@skku.edu (W.-Y.P.); 2Department of Health Sciences and Technology, Samsung Advanced Institute for Health Sciences & Technology, Sungkyunkwan University, Seoul 06355, Republic of Korea; 3Department of Breast Cancer Center, Samsung Medical Center, Seoul 06351, Republic of Korea; sangmin3005.kim@samsung.com; 4Department of Surgery, Samsung Medical Center, Sungkyunkwan University School of Medicine, Seoul 06351, Republic of Korea; 5Planit Healthcare Inc., Seoul 06235, Republic of Korea; eastwise37@gmail.com; 6Department of Digital Health, Samsung Advanced Institute for Health Sciences & Technology, Sungkyunkwan University, Seoul 06355, Republic of Korea; leets1984@gmail.com; 7Department of Molecular Cell Biology, Sungkyunkwan University School of Medicine, Suwon 16419, Republic of Korea

**Keywords:** scRNA-seq, SIDR-seq, transcriptome, CNV alteration, breast cancer, tumor microenvironment

## Abstract

Hormone receptor-positive and HER2-negative breast cancer (HR+/HER2-BC) is the most common type with a favorable prognosis under endocrine therapy. However, it still demonstrates unpredictable progression and recurrences influenced by high tumoral diversity and microenvironmental status. To address these heterogeneous molecular characteristics of HR+/HER2-BC, we aimed to simultaneously characterize its transcriptomic landscape and genetic architecture at the same resolution. Using advanced single-cell RNA and DNA sequencing techniques together, we defined four distinct tumor subtypes. Notably, the migratory tumor subtype was closely linked to genomic alterations of EGFR, related to the tumor-promoting behavior of IL6-positive inflammatory tumor-associated fibroblast, and contributing to poor prognosis. Our study comprehensively utilizes integrated analysis to uncover the complex dynamics of this breast cancer subtype, highlighting the pivotal role of the migratory tumor subtype in influencing surrounding cells. This sheds light on potential therapeutic targets by offering enhanced insights for HR+/HER2-BC treatment.

## 1. Introduction

Breast cancer is the most common cancer in women worldwide and the second most prominent cause of cancer-related mortality in this population [1]. Human breast cancers are highly heterogeneous, with up to ten described subtypes [2,3]. Among these subtypes, hormone receptor-positive and HER2-negative breast cancer (HR+/HER2-BC) is the most heterogeneous subtype and is clinically known to have a better prognosis than other molecular subtypes [4,5]. HR+/HER2-BC includes luminal A and luminal B subtypes, characterized by their distinct gene expression profiles. Luminal A tumors typically express higher levels of estrogen receptor (ER)-related genes and have lower proliferation rates, whereas luminal B tumors show higher proliferation rates and lower expression of ER-related genes, often requiring more aggressive treatment [6,7]. In addition to intertumoral heterogeneity, significant efforts have been made to characterize intratumoral heterogeneity in a clinically relevant manner. However, gaps in our understanding of the full spectrum of cellular heterogeneity in breast cancer cells challenge the investigation of cancer initiation and progression, which are associated with patient clinical outcomes.

Combining single-cell genomics and gene expression analysis can reveal both the qualitative and quantitative characteristics of cancer development and gene regulation [8,9]. Simple and efficient methods for the simultaneous isolation of genomic DNA and total RNA from single cells, such as G&T-seq, and DR-seq, allow for the integrated analysis of genomic features and RNA expression profiles from the same individual cell, enabling a comprehensive understanding of cellular heterogeneity and complexity at the single-cell level [10,11,12]. 

In this study, we aimed to demonstrate the efficient integration analysis of two complementary single-cell methods. Droplet single-cell RNA sequencing (scRNA-seq) was used to evaluate the heterogeneity of tumors and their microenvironmental cells from a large population of cells, and simultaneous isolation of genomic DNA and total RNA sequencing (SIDR-seq) was used for the linkage between transcriptomic characteristics and genetic alterations. 

Our primary objective was to identify a distinct tumor cell type that plays a significant role in the behavior or dynamics of this type of breast cancer, when considering both transcriptomic and genetic features. Additionally, we aimed to elucidate the collaborative interactions of this cell type with the tumor microenvironment.

## 2. Results

### 2.1. Integrated Transcriptomic Profiling of HR+/HER2-Breast Cancer

We obtained tumor tissue from 14 patients with HR+/HER2-BC who were not treated with neoadjuvant chemotherapy. ER/PR/HER2 status was assessed using clinical and histological criteria (Appendix A). We performed scRNA-seq using a 10× Chromium platform on nine patients and obtained 24,803 cells in total (Figure 1a). We also used SIDR-seq on 14 tumor tissues sorted by surface EpCAM. A total of 228 SIDR-WGS and 408 SIDR-WTS cells were captured, and their data quality was confirmed by comparison with bulk RNA and DNA sequencing data (Appendix A).

UMAP of 10x scRNA-seq identified six major cell types based on unsupervised clustering and canonical marker gene expression across individuals (Figure 1b,c). Epithelial cells were discriminated by *EPCAM*, the general epithelial marker. Most of the epithelial cells expressed luminal cell type markers, including *KRT18*, *KRT8* and *KRT19*. Additionally, partial expression of luminal differentiation-related markers such as *PGR*, *ESR1*, *AR* and *PRLR* were observed. Myoepithelial cells were distinguished from epithelial cell types by their exclusive expression of the myoepithelial markers *TAGLN*, *KRT5*, *KRT14* and *KRT17*.

The tumor microenvironment (TME) of HR+/HER2- breast cancer cells is composed of *CD3D*+ T cells, *CD68*+ macrophages, *COL1A1*+ fibroblasts, and *FLT1*+ endothelial cells. The scRNA-seq cell type proportion of the nine patients showed a similar tendency to the estimated stromal and immune statuses from the bulk RNA-seq analysis for patients with high immune and stromal scores in bulk RNA-seq and exhibited a corresponding increase in the percentage of T-cells and fibroblast cells in the scRNA-seq analysis (Figure 1d,e). We compared the result of PAM50 classification to identify molecular subtypes of human breast cancer [13]. The PAM50 results using bulk RNA-seq were two luminal A, six luminal B, and one Her2 PAM50 subtype, which was consistent with the distribution of IHC-based HR+/HER2-BC reported in other studies [14,15]. In this study, patients with high expression of hormone related genes, such as *ESR1*, *PGR* and *AR* in bulk RNA-seq, had a comparably greater proportion of epithelial cells in scRNA-seq. However, a patient with the highest proportion of non-epithelial cells in scRNA-seq was classified as Her2 PAM50 subtype in bulk RNA-seq, indicating the potential impact of cell type composition on tumor characteristics. The study highlights the significant transcriptomic variation observed in HR+/HER2-BC patients.

### 2.2. scRNA-seq Reveals a Migratory Functional Tumor Subtype of HR+/HER2-Breast Cancer

To delineate the heterogeneous characteristics of HR+/HER2-BC tumor epithelial cells, we identified a subset of 17,990 tumor epithelial cells and classified them into four single-cell-defined functional (SC-f) tumor subtypes based on their functional roles (Figure 2a). Of the nine individuals, three were migratory tumor subtype-dominant and six as secretory SC-f tumor subtype-dominant (Appendix A).

We inferred the CNV ploidy status of the epithelial cells and confirmed the tumor malignancy among four SC-f subtypes (Figure 2b). In the dysfunctional subtype, there were relatively low aneuploid cells compared to other subtypes. Within malignant tumor subtypes, breast cancer-related CNV signatures, a gain of 1q, 16p and 17q, and a loss of 16q were detected in all subtypes compared to normal breast epithelial cells (Appendix A). In HR+/HER2-BC, which is primarily composed of luminal tissue types, we assessed luminal cell markers across four SC-f subtypes. We noted a relative upregulation of specific luminal markers, *KRT8* and *KRT7*, as well as distinctive expression of basal-like marker *KRT81* [16] in the migratory tumor subtype. In addition, *CDKN1B* (encoding p27), a known stem and progenitor marker [17], exhibited exclusive expression within the migratory tumor subtype. This finding suggests a unique molecular signature of the migratory subtype, possibly an indication of an intermediate status between the luminal and basal phenotypes, contributing to their migratory behavior in HR+/HER2-BC. Conversely, the hormone-related marker *ESR1* was expressed in secretory and proliferating subtypes, indicating their mature forms, while *MKI67*, a proliferating marker, was uniquely expressed in the proliferating subtype (Figure 2c). The SC-f subtype-specific differentially expressed genes (DEGs) exhibited transcriptional distinctions among tumor epithelial cells (Figure 2d and Appendix A). In particular, the migratory subtype was distinguished by *S100A10*, *ANGPTL4*, *TM4SF1*, *SFN* and *AREG*, which were associated with tissue migration and angiogenesis, as well as *CXCL8* and *TNFRSF12A*, which contribute to the immune system. Gene ontology (GO) analysis revealed the functional role of each SC-f tumor subtype. The migratory subtype may contribute to epithelial cell migration through angiogenesis and the interferon–gamma pathway. The secretory subtype might mainly relate to hormone generation, secretion, and gland cell maintenance. The proliferating subtype is solely focused on cell cycle elaboration for increasing cell numbers. Lastly, the dysfunctional subtype, defined by its lack of clear involvement in specific functions, appears to have a minor role, suggesting limited functional activity among the four tumor subtypes (Figure 2e and Appendix A).

Next, we compared the cluster characteristics with those of the PAM50 intrinsic subtype classification (Figure 2f). The secretory and proliferating subtypes were representative for Luminal A and Luminal B classification, respectively. On the other hand, the migratory subtype showed shared characteristics with Luminal A and Normal-like classification, as they had low expression of known breast cancer markers such as *ESR1*, *PGR* and *ERBB2* (Appendix A). HR+/HER2-BC is highly diverse, reflecting various PAM50 classification results. Defining the characteristics of the migratory and dysfunctional subtypes using conventional methods is particularly challenging. In addition, the correlation between the four SC-f tumor subtypes indicated that the migratory subtype primarily exhibited distinct characteristics (Figure 2g). 

Gene set enrichment analysis (GSEA) results also showed that epithelial–mesenchymal transition (EMT), hypoxia and tumor necrosis factor-α (TNFα) signaling were exclusively enriched, whereas estrogen responses were negatively enriched (Figure 2h and Appendix A). We observed a significant increase in the expression of cancer stem cell markers, including *CD44*, *VEGFA*, *ITGB1*, *ALDH1A3*, *SOX9* and *NECTIN4* [18,19], and enrichment in stemness-related pathways, such as the hedgehog, PI3K-AKT-mTOR, and WNT signaling pathways (Figure 2i and Appendix A). The trajectory revealed that the migratory subtype was in a different pseudotime state from the other cell types and the secretory and proliferating subtypes were in the same branch, whereas the dysfunctional subtype went through a separate branch (Figure 2j and Appendix A). In addition, it indicated that the migratory subtype was the most immature form [20] of luminal cell and possessed greater potential function, such as migration activity.

### 2.3. SIDR-seq Elucidates the Effects of Genomic Aberrations on Migratory Subtype That Increase the Risk of Early Recurrence of HR+/HER2-Breast Cancer

Because SIDR-seq is an integrative system [10], we were able to study how the migratory subtype affects patient prognosis when paired with genomic changes. First, 408 SIDR-WTS cells from 14 patients were clustered using SNN neighboring and assigned to each cluster based on the SC-f tumor subtype signature from 10x scRNA-seq (Figure 3a and Appendix A). All captured SIDR-WTS were epithelial cells and reproduced the SC-f tumor subtypes and known HR+ breast cancer signatures as reported by Doane et al. [21], migration and EMT (Figure 3b). The dysfunctional subtype was matched with a cluster in which the signatures of the other three subtypes were absent. This was because the dysfunctional subtype did not show any specific genetic features distinct from those of the other three groups. Then, we embedded the genomic information from SIDR-WGS data into the same UMAP to investigate the genomic alterations in each SC tumor group. The overall frequency of CNV abnormalities was higher in the proliferating and secretory subtypes than the migratory and dysfunctional subtype (Figure 3c). Furthermore, these results were consistent with those from the scRNA-seq analysis (Appendix A) and the trajectory analysis by scRNA-seq data (Appendix A), which indicated that after the emergence of the migratory subtype, additional CNV accumulation occurred and led to the development of the secretory or proliferating subtypes in HR+/HER2-BC. We found that each patient exhibited unique genetic characteristics (Appendix A); however, when they were integrated into the UMAP of the SIDR-WTS, a unique CNV pattern was observed for each SC-f tumor subtype (Appendix A). 

When we analyzed the focal CNVs of the four SC-f subtypes separately, most major changes were observed at higher frequencies in other subtypes than the migratory subtype. However, some did show the highest frequency in the migratory subtype, including gains of 7p11.2, 7p22.3 and 7q36.1, and losses of 16q23.1, 16q24.2, 16q12.1 and 16p11.2 (Figure 3d and Appendix A). To further investigate the prognostic implications of CNVs in the migratory SC-f tumor subtype, we tested the *EGFR* (7p11.2) gains with a migratory tumor signature in HR+/HER2-BC using the TCGA-BRCA (Figure 3e). Patients with both a high migratory signature and *EGFR* gains showed poorer disease-free survival (DFS) than those who had a migratory signature only. However, the presence of *EGFR* gains did not impact patients with low migratory features. This indicates that the combination of a high migratory signature and *EGFR* gains may be a prognostic factor for poor outcome in breast cancer patients. However, the clinical relevance of other migratory specific CNV regions was found to lack validity (Appendix A). Moreover, we identified 10 transcriptomic targets with clinical implications, particularly when co-existing with *EGFR* gain (Figure 3f). Among these genes, none of them showed any significant prognostic impact without *EGFR* gain. However, the hazard ratio for these genes increased significantly with the presence of *EGFR* gain. Notably, *SPINK1* was found to have the most significant impact on the DFS of breast cancer patients with *EGFR* gain. We strongly suggest that tumors with the same transcriptomic characteristics could have different clinical prognoses depending on their genetic changes, and developing integrated strategies to identify prognostic factors could be a practical approach for improving patient outcomes.

We have reproduced the classification of SC-f tumor subtypes using bulk RNA-seq data (Appendix A). Fourteen individuals were categorized into two groups: one with a migratory subtype signature and the other with a non-migratory signature. Genes related to extracellular organization, angiogenesis and tissue migration were upregulated in the migratory subtype dominant group (Appendix A). These findings were consistent with the results of the scRNA-seq analysis.

### 2.4. IL6 Expressing iCAF Is Highly Activated in HR+/HER2-Breast Cancer

To investigate fibroblasts in the TME, we captured 1192 scRNA-seq cells, performed re-clustering, and annotated them as three types of cancer-associated fibroblasts (CAFs): two inflammatory CAFs (iCAFs) and one subset of myofibroblast-like CAF (myCAF) (Figure 4a and Appendix A). The iCAFs of HR+/HER2-BC expressed pan-CAF markers such as *PDPN*, *POSTN*, *DCN*, *FBLN1* and *FAP*, whereas, myCAF showed high levels of *ACTA2*, *RGS5*, *MCAM*, *MYLK* and *PPP1R14A* (Figure 4b). Among iCAFs, the IL6^high^ iCAF distinctly expressed *IL6*, *IL32* and *CXCL8*, and the CXCL12^high^ iCAF had relatively high expression of *CXCL12* and *CXCL14*. Numerous oncogenic pathways were upregulated in IL6^high^ iCAF compared to other CAFs (Figure 4c). This indicated that IL6^high^ iCAF was significantly activated in fibroblast cells and might be associated with the TME through WNT, VEGF, cytokines and metabolic pathways. The trajectory between the two iCAFs clearly showed that they were located at opposite sites of pseudotimes, with different biases in gene expression (Figure 4d,e). The IL6^high^ iCAF was enriched in cytokines including *IL33*, *CXCL1*, *CXCL3* and *CCL3L3*, and surface molecules such as *ITGA1*, *ICAM1*, *CD44*, *CD59* and *TNFRSF12A*. In contrast, matrix molecules such as *TALN*, *PDGFRL* and *POSN*, and the induced genes *IFITM2*, *IFITM3* and *IFI27* were polarized (Appendix A). To demonstrate the clinical impact of each CAF, we performed an overall survival analysis according to the three types of CAFs in HR+/HER2-BC using TCGA-BRCA [22] and METABRIC [23] data (Figure 4f,g). Patients who were dominant for IL6^high^ iCAF and myCAF showed worse survival in both cohorts; however, those with high CXCL12^high^ iCAF levels had a better prognosis. These results showed a consistent trend in both cohorts, with statistically significant findings observed in METABRIC, which had a larger number of patients. As a result, the two iCAFs had different functions and affected the prognosis of patients with HR+/HER2-BC, suggesting that IL6^high^ iCAF has more diverse features and contributes to the conversion of the TME into a tumorigenic environment.

### 2.5. Diverse Role of Migratory HR+/HER2-BC Interactions with Multiple CAFs

We analyzed the association network between three major SC-f tumor subtypes and three CAF subtypes to examine how cells interact to achieve the same tumorigenic process. A network map using an enriched ontology pathway showed that only the migratory subtype shared pathways with all the CAFs, and each CAF combination was involved in different tumorigenic processes (Figure 5a and Appendix A). When migratory tumor cells overlapped with myCAF, they mainly promoted cell migration and adhesion via the PI3-Akt signaling pathway. In combination with IL6^high^ iCAF and CXCL12^high^, it was involved in cancer immune reactions through the TNF, NF-kappa B and IL-17 pathways. Migratory tumor cells and IL6^high^ iCAF were closely related to cancer metabolism, such as carbon and amino acid processing by HIF-1 signaling and activated cell cycle pathways. This suggests that the migratory subtype may act as a hub that helps and regulates the function of CAFs in HR+/HER2-BC.

To investigate the tumor microenvironmental characteristics according to the level of SC-f subtype, T-cells and macrophage cells were also classified into subtypes based on canonical markers and M1/M2 signatures (Appendix A). The proportion of the TME subtypes was compared by ratio of observed to expected cell numbers (Ro/e). As a result, four patients (BC_03, 06, 14 and 15) were selected as the migratory SC-f high group. They also showed an increased IL6^high^ iCAF, M1-like macrophage and CD4+ effector memory T cell (Tem) fraction (Appendix A). 

Comparison of fibroblast cells between the two groups revealed upregulation of genes associated with the pathways depicted in Figure 5a, specifically in the migratory SC-f subtype high (Mi-h) group (Figure 5b). *FGFR1*, *VEGFA*, *LAMA4*, *ANGPT2* and *LPAR6* were members of the PI3-Akt signaling pathway, while *CEBPB*, *NFKBIA*, *TNFAIP3*, *CXCL8*, *FOSL1*, *IL6*, *ICAM1* and *LIF* were member of the TNF, NF-kappa B and IL-17 pathways, and *ENO1*, *PGAM1*, *PKM*, *TPI1*, *LDHA* and *SERPINE1* were member of the HIF-1 signaling and biosynthesis pathways. Additionally, most of the functional genes in Figure 5b were highly expressed in IL6^high^ iCAF, compared to other CAFs (Appendix A). These findings support the hypothesis that IL6^high^ iCAF may be a key player in promoting tumor progression in patients with the migratory SC-f subtype.

Finally, we investigated the ligand–receptor interactions between tumor cells and cells in the tumor microenvironment. We revealed that IL6^high^ iCAF showed a higher frequency of interaction with other cell types, specifically with endothelial cells, CXCL12^high^ iCAF, myCAF and the migratory SC-f subtype (Figure 5c). Next, we compared the paired interactions in four types of SC-f tumor subtypes to discover the migratory subtype-specific interactions (Appendix A). We found that *AREG_EGFR* interaction exclusively occurred between migratory tumor and IL6^high^ iCAF. IL6^high^ iCAF exhibited a distinct ligand expression involving *IL6*, which interacted with migratory tumor and M1/M2-like TAMs through *IL6R*, and *HRH1* in endothelial cells to activate the inflammatory response and angiogenesis [24]. CXCL12^high^ iCAF was also associated in tumor growth by *FGF1/7/10* and *CXCL12* interactions (Figure 5d). Our analysis of potential ligand–receptor interactions revealed that the migratory subtype interacts with multiple cell types, including IL6^high^ iCAF, CXCL12^high^ iCAF, myCAF, M1/M2-like TAMs and endothelial cells. The interaction between the migratory subtype and IL6^high^ iCAF was the most specific and EGFR-related, which may indicate its importance in promoting tumor progression and mesenchymal transition for cancer progression. 

## 3. Discussion

We used two complementary single-cell sequencing approaches to evoke new insights into the heterogeneity of HR+/HER2-BC. 10x scRNA-seq provided high-resolution gene expression datasets of both tumor and TME cells and SIDR-seq simultaneously profiled both the transcriptional and genomic features of the same individual tumor cells [10]. 

We suggested four novel SC-f tumor subtypes found in HR+/HER2-BC tumors, based on their genomic changes and functional roles. Among them, we focused on the migratory subtype with both luminal and basal epithelial phenotypic features, low hormone receptor expression, and comparably lower accumulation of CNV. Tumor cells found in breast cancer expressing high *KRT7* and low *ESR1* phenotypes may experience increased plasticity and heterogeneity [25]. This could potentially lead to the acquisition of certain functions associated with basal cell types, such as migration or stemness [26,27]. The migratory subtype also exhibits similar characteristics to those of cancer stem cells (CSC) with expression of known CSC markers (*CD44*, *SOX9*, *ALDH1A3* and *VEGFA*) and those for tissue migration, angiogenesis, EMT and the TNFA pathway [28,29,30]. The production of *CXCL8* enhanced the activity of CSCs by activating CXCR1/2 receptors in both an EGFR/HER2-dependent and independent manner [31]. The migratory subtype in this study also showed a highly biased expression of *CXCL8* and other stemness factors, such as *SOD2* and *FGFR1* [32,33].

Our analysis identified specific marker genes for the migratory SC-f tumor subtype, as well as sub cell types of CAFs. Appendix A summarizes the top marker genes for each identified subtype, originating from two complementary methods: 10x scRNA-seq and SIDR-seq. This comprehensive marker gene list provides valuable insights into the molecular characteristics and potential functional roles of these subtypes in breast cancer progression.

Genetic information of four SC-f tumor subtypes suggests that specific stimuli may trigger genetic instability in the migratory subtype that leads to the emergence of different malignant cell subtypes, such as the secretory and proliferating subtypes, and ultimately increased tumor heterogeneity. Studies have shown that genomic alterations in *EGFR* play an important role in tumor cell growth in breast cancer [34,35,36]. In this study, we observed that 35% of the migratory subtype showed an *EGFR* gain on chromosome 7p11.2 in SIDR-seq. Similarly, in TCGA-BRCA, approximately 28% of patients in the high migratory subtype group had an *EGFR* gain. These patients had shorter recurrence periods than those with *EGFR* loss and no *EGFR* alteration [37,38,39]. 

The interaction between tumor and TME cells may play a critical role in regulating cancer cell behavior [40,41,42]. Our study emphasized the role of IL6^high^ CAF on tumor cells for elevating level of various cytokine genes including *IL6*, *IL32*, *IL33*, *CXCL1*, *CXCL3* and *CXCL8*, and oncogenic pathways, such as the Wnt and VEGF signaling pathways. Studies have shown that IL6-secreting CAFs can promote tamoxifen resistance through the activation of the JAK/STAT3 or PI3K/AKT pathways, induce EMT [43] and enhance the acquisition of invasive and metastatic properties. 

Moreover, our findings highlight the critical cooperation between migratory tumor cells and CAFs, particularly iCAFs. Genetic alterations in migratory tumor cells, such as *EGFR* gains, might enhance their aggressive phenotype and foster intricate cell–cell interactions with iCAFs. These interactions contribute to increased tumor heterogeneity and promote the secretion of pro-tumorigenic cytokines like IL-6, creating an immunosuppressive microenvironment. Understanding this bidirectional communication provides valuable insights for developing strategies to disrupt the mechanisms underlying tumor heterogeneity and immune evasion in HR+/HER2-BC.

Our findings suggest several potential practical applications for targeting the migratory SC-f tumor subtype and its interactions with the TME. Since a significant portion of migratory subtype tumors exhibit EGFR gains, therapies targeting *EGFR*, such as tyrosine kinase inhibitors (TKIs) could be effective in this patient subset, as is currently being investigated for triple-negative breast cancer (TNBC) patients [44,45]. Targeting CSC-associated pathways exhibited in the migratory subtype could reduce tumor heterogeneity and prevent early recurrence [46]. Further, our study indicates that IL-6-secreting CAFs play a crucial role in promoting tumor progression. Targeting the IL-6 mediated JAK/STAT3 or PI3K/AKT pathways could potentially disrupt this interaction and improve clinical outcomes for HR+/HER2-BC [47].

## 4. Materials and Methods

### 4.1. Patient Recruitment and Tumor Samples

A total of 14 patients diagnosed with HR+/HER2-BC were recruited for this study. Tissue specimens were obtained by surgical excision without prior treatment. This study was approved by the Institutional Review Board (IRB) of Samsung Medical Center, and all patients provided signed informed consent for collection of specimens and detailed analyses of the derived genetic materials (Institutional Review Board no. 2015-12-094). Tumor tissues were mechanically dissociated and enzymatically digested on the day of the surgery. Dead cells were removed using Ficoll-Paque PLUS (GE Healthcare, Uppsala, Sweden).

### 4.2. Sequencing Procedure for Bulk and SIDR-seq

Isolation of genomic DNA and total RNA was performed according to previously reported protocols [10]. RNA samples were reverse transcribed and pre-amplified using SMART-Seq2, following the manufacturer’s protocol (SMARTer Ultra Low Input RNA for Sequencing-v3; Clontech, Mountain View, CA, USA). Using 1-ng aliquots of each cDNA sample, a WTS library was prepared using a Nextera XT DNA Sample Prep Kit (Illumina, San Diego, CA, USA), following the manufacturer’s instructions. The libraries were sequenced on a HiSeq 2500 system using 100-bp paired-end sequencing. Whole-genome amplification was performed according to the manufacturer’s protocol (Repli-G single-cell kit; Qiagen, Hilden, Germany). The WGS libraries were constructed using a TruSeq Nano DNA Library Prep Kit (Illumina, San Diego, CA, USA) according to the protocol for multiplexed paired-end sequencing sample preparation. Low-coverage genome sequencing was performed on an Illumina HiSeq 2500 system with 100-bp paired-end sequencing. 

### 4.3. Sequencing Procedure for 10x scRNA-seq

Library preparation was performed using the Chromium Single Cell 3’ Gene Expression system (version 2) and processed following the manufacturer’s protocol (10x Genomics, Pleasanton, CA, USA). The library was sequenced on a HiSeq 2500 system using 100-bp paired-end sequencing.

### 4.4. DNA Sequencing Processing for Bulk and SIDR-seq

WGS reads from bulk-seq and SIDR-seq were aligned to the human reference genome (version hg19) using Bowtie 2 (version 2.2.9) [48] with the default parameters. Putative duplicates among the mapped reads were marked using the MarkDuplicate module of the Picard tool (version 2.17.5; http://broadinstitute.github.io/picard, accessed on 27 July 2019). The read counts per base were quantified using Bedtools (version 2.17.0). To assess data quality, we first calculated the index of dispersion and then cells with an index of dispersion lower than 1.25 were excluded. The genomic copy number was estimated using Ginkgo (https://github.com/robertaboukhalil/ginkgo, accessed on 25 November 2019). The median genomic length of each bin was 100 kb. Sex chromosomes were excluded from downstream analysis. 

### 4.5. RNA Sequencing Processing for Bulk and SIDR-seq

WTS reads from bulk and SIDR-seq were aligned to the human reference genome (version hg38) using the 2-pass default mode of STAR Aligner (version 2.5) with Gencode annotation (version 34). Gene expression was quantified in units of transcripts per million (TPM) using RSEM (version 1.3.0) with default parameters. For WTS from SIDR-seq (SIDR-WTS), genes that were detected in fewer than three cells were filtered out and only cells that expressed more than 200 features were used. After pre-filtering, normalization and scaling were performed using the Seurat R package (version 4.0.5).

### 4.6. RNA Sequencing Processing and Analysis for 10x scRNA-seq

Raw sequencing data were processed using Cell Ranger (version 4) using the default setting (https://support.10xgenomics.com/single-cell-gene-expression/software/pipelines/latest/what-is-cell-ranger, accessed on 1 March 2021). The reads were aligned to the human reference genome (hg38). The Seurat R package (version 4.0.5) was used to normalize expression values. Genes that were detected in less than three cells were filtered out; the total number of expressed genes/cells was 200 < nGenes < 6000, and less than 25% of the UMIs mapped to the mitochondria-originated genes were used. Dimensionality was reduced through principal component analysis (PCA) using Seurat’s pipeline [49] and batch corrected using Harmony [50]. 

### 4.7. Copy Number Variation (CNV) Inference with scRNA-seq

We applied CNV inference to identify tumor epithelial cells within droplet-based scRNA-seq cells. The CopyKAT R package [51] was performed with default parameters using all count matrices. For comparison with normal breast epithelial cells, we ran the inferCNV R package [52] with i6 HMM prediction (inferCNV of the Trinity CTAT Project, https://github.com/broadinstitute/inferCNV, accessed on 18 March 2021) using epithelial cells, including myoepithelial cells from this study and normal breast epithelial cells from GSE113196 [2] as a normal control.

### 4.8. PAM50 Intrinsic Molecule Subtype Classification

The bulk-RNA-seq and scRNA-seq data were assigned to five PAM50 intrinsic subtypes: Luminal A (LumA), Luminal B (LumB), human epidermal growth factor receptor-2 positive (HER2), basal-like (basal), and normal-like (normal), which were analyzed using the PAM50Preds function of the Genefu R package [53] with a normalized matrix. This utilizes Gaussian mixture models for assessing probability of PAM50 classification of each piece of data [54].

### 4.9. Identification of Differentially Expressed Genes and Gene Set Enrichment Analysis

DEGs for each cluster were calculated using the FindMarker function of the Seurat package and MAST [55]. These genes were used for the following functional analyses: GO, KEGG, and HALLMARKER pathway enrichment within epithelial and fibroblast cells. We used the clusterProfiler (version 4.6.0) R package [56] to compare the biological characteristics between sub-clusters (compareCluster function).

### 4.10. Trajectory Pseudotime Analysis

To infer single-cell trajectories for epithelial and fibroblast cells, DDRTree in the Monocle (version 2.26.0) R package [57] was used. Transcriptomic data were normalized with the Seurat R package and used directly as inputs for Monocle. 

### 4.11. Segmentation of Copy Number Data from SIDR-WGS

To detect the genome location with copy number aberrations (CNA) from preprocessed whole genome sequencing data from SIDR-seq (SIDR-WGS), we used the copynumber R package [58] for piecewise constant fit (PCF) segmentation with visualization. This provided the frequency of gains or losses at a genomic position across a grouped sample.

### 4.12. Data Collection and Statistics for Survival Analysis

The clinical data, gene expression profiles, and CNAs of the TCGA-BRCA and METABRIC cohorts were collected from the cBioPortal database (http://www.cbioportal.org/, accessed on 25 August 2021). Cox proportional hazard (Coxph) regression analyses were performed to assess the significance of the relationships between transcriptomic/genetic profiles and recurrence-free/overall survival in HR+/HER2-BC. The Survminer R package was used to determine the optimal cut-off points for categorizing the migratory subtype and the three types of CAFs using the surv_cutpoint and surv_categorize functions. Additionally, the Survfit function was used to estimate the Kaplan–Meier (KM) curves for clinical outcomes (https://rpkgs.datanovia.com/survminer/, accessed on 6 January 2022). When analyzing using a two-factor KM analysis with transcriptomic and genetic features as the same time, for visual clarity, we split the data into two plots based on transcriptomic features.

### 4.13. Cell Type Estimation in Bulk RNA-seq Using scRNA-seq-Based References

The proportions of tumor and CAF subtypes from bulk RNA-seq data (TCGA-BRCA and METABRIC) were estimated using CIBERSORTx (https://cibersortx.stanford.edu, accessed on 22 November 2022) [59]. Subtype-annotated count data from scRNA-seq of epithelial and fibroblast cells were used as references to build a custom signature matrix. We performed CIBERSORTx using custom references with a quantile normalization algorithm.

### 4.14. Cell-Cell Interaction Analysis Using CellPhoneDB

We applied the well-established CellPhoneDB (version 2.0.0) method to study cell–cell communication across all the cell types. CellPhoneDB provides a repository of curated receptors, ligands, and their interactions [60]. Pairwise comparisons between all cell types yielded the absolute number of matched ligand–receptor pairs, mean of the average expression levels of the receptor and ligand molecules, and *p*-values for the likelihood of specificity in the corresponding two cell types.

## 5. Conclusions

Our study found that patients with migratory phenotype tumors exhibited a prevailing chromosome 7p gain (locus for *EGFR*) and showed earlier relapses, according to the publicly available dataset. We further extend this finding by elucidating the functional role of migratory tumor cells as a key modulator of different types of CAFs: IL6+ iCAF, CXCL12+ iCAF and myCAF. This orchestration contributes to creating a tumor-friendly environment and leads to poor prognosis, including recurrence in HR+/HER2-BC.

However, the limitation of this study is its inability to directly establish migratory tumor prognosis due to its small sample size and short follow-up. Additionally, distinct tumor properties, including migration and stemness, were not definitively confirmed experimentally. Thus, larger studies and advanced techniques are needed to fully understand the role of migratory phenotype tumor cells in HR+/HER2-BC and their interactions with TME cells.

Nevertheless, our study highlights the effectiveness of simultaneous genomic and transcriptomic profiles of HR+/HER2-BC at the single-cell level, with particular focus on the migratory SC-f tumor subtype in relation to *EGFR* genomic aberration and the interplay between CAFs. Further research in this area is essential to advance our understanding of the heterogeneity of hormone receptor-positive breast cancer and develop more effective therapeutic targets for patients. 

## Figures and Tables

**Figure 1 ijms-25-05876-f001:**
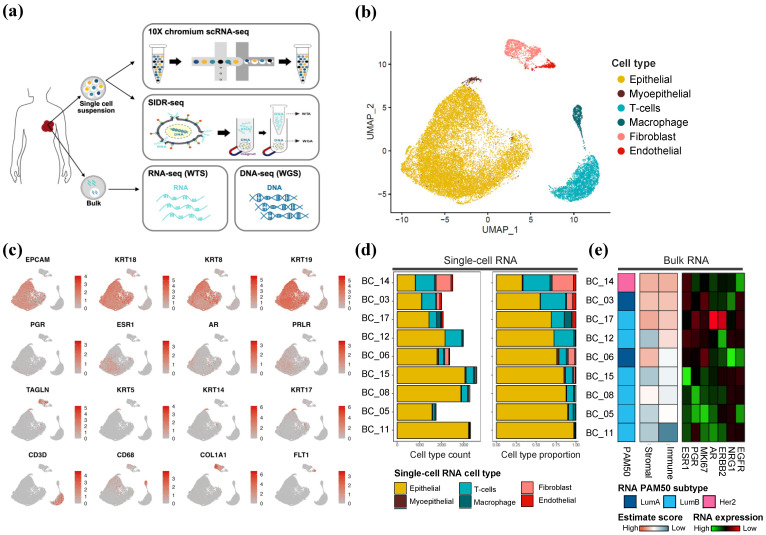
Integrated transcriptomic profiling of HR+/HER2-BC. (**a**) Overview of multi-omics library construction according to bulk and single-cell levels. (**b**) UMAP plot of 24,803 scRNA-seq cells from nine HR+/HER2-BC patients annotated with six major cell types. (**c**) Expression of cell type specific genes and known breast cancer biomarkers. (**d**) Cell type proportion HR+/HER2-BC by individual. (**e**) Transcriptomic profile using bulk RNA-seq showing PAM50 subtype (**left**), tumor infiltrating scores by “ESTIMATE” (**middle**), and expression of genes from PAM50 discrimination (**right**).

**Figure 2 ijms-25-05876-f002:**
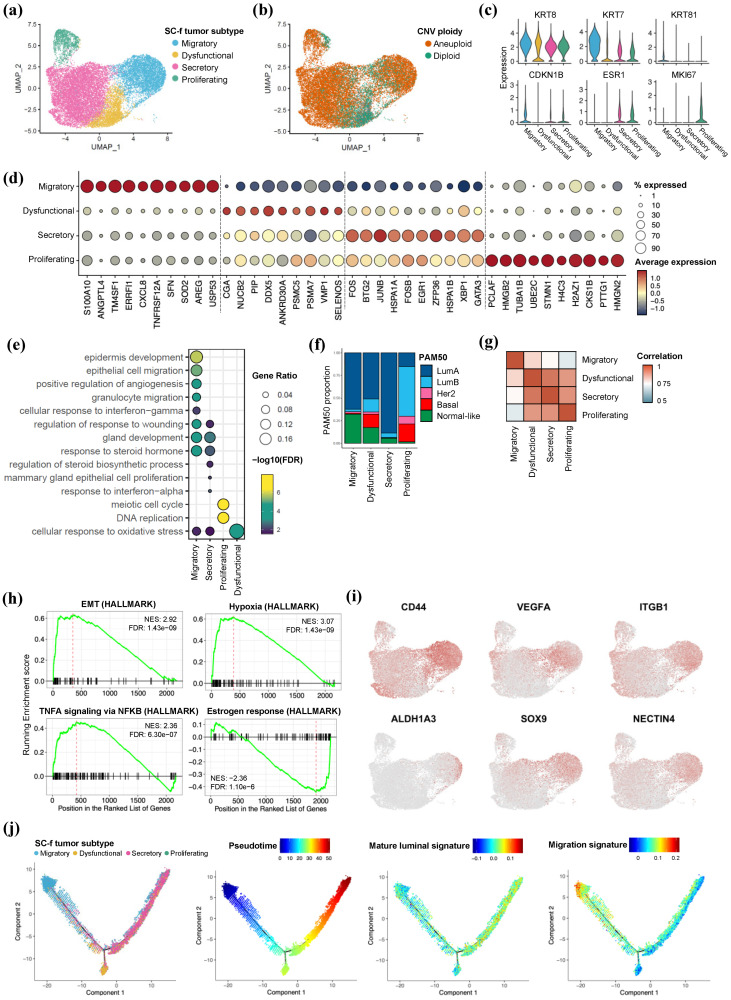
scRNA-seq revealed migratory SC-f tumor subtype of HR+/HER2-BC. (**a**) UMAP plot of 17,990 epithelial cells annotated with sub cell types. (**b**) Copy number ploidy of tumor epithelial cell inferred by CopyKAT. (**c**) Expression of functional markers of luminal cell across four SC-f tumor subtypes. (**d**) Top 10 DEGs of each SC-f tumor subtype. (**e**) GO pathway analysis using up-regulated genes in each SC-f tumor subtype. (**f**) Comparison of SC-f tumor subtype classification and intrinsic molecular subtype PAM50 results. (**g**) Correlations with variable genes between four SC-f tumor subtypes. (**h**) Running enrichment score (ES) of Hallmark signatures of migratory subtype. Green line represents the running ES, red vertical line indicates the position of the peak ES, marking the point of maximum enrichment and genes contributing most to the ES, shown up to the peak ES on the x-axis. (**i**) Expression of known biomarkers for cancer stemness. (**j**) Trajectory showing inferred pseudotime within SC-f tumor subtypes. Module score of mature luminal signature and epithelial cell migration signature was projected in the trajectory dimension.

**Figure 3 ijms-25-05876-f003:**
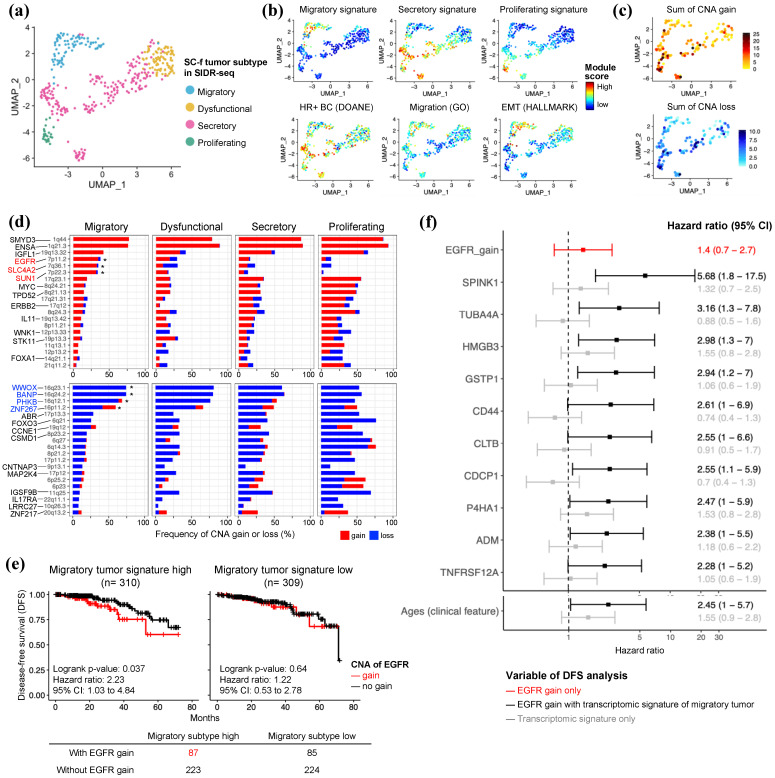
Concurrent genomic aberrations in the migratory subtype contribute to early recurrence in HR+/HER2-BC. (**a**) UMAP plot of SIDR-WTS annotated by SC-f tumor subtype classification. (**b**) Module score calculated with three types of SC-f tumor subtype signatures, hormone positive breast cancer signature, migration and EMT signature in SIDR-WTS. (**c**) Aggregation of detected CNAs across entire chromosomes in SIDR-WGS data. (**d**) Frequency of CNA of oncogenes located in each chromosomal regions (* indicates the migratory subtype specific gain or loss). Genes corresponding to each region were marked with representative genes suggested by TCGA-BRCA. (**e**) Two-factor Kaplan–Meier survival analysis conducted on 619 hormone receptor positive TCGA-BRCA patients with CNA gain and the level of migratory tumor subtype signature score. (**f**) Forest plot of DFS by EGFR gain and migratory tumor subtype signature gene expression. Hazard ratio was estimated based on a Cox proportional hazards model for EGFR gain with high expression of migratory tumor subtype signature gene and high expression of migratory tumor subtype signature only, without EGFR gain effect. Age was incorporated as a covariate to assess the effects of other variables. A 95% confidence interval (CI) is represented by a horizontal line and all CI is below this.

**Figure 4 ijms-25-05876-f004:**
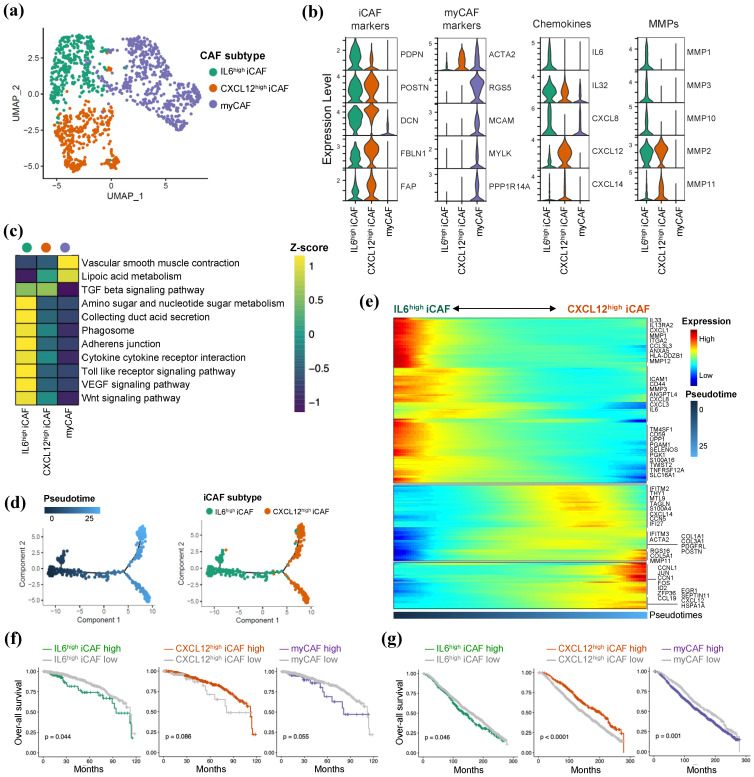
IL6 secreted iCAF is highly activated in HR+/HER2-BC. (**a**) UMAP plot of 1192 CAFs of HR+/HER2-BC representing four subpopulations from scRNA-seq. (**b**) Expression of two types of CAF markers, chemokines/cytokines and MMPs in HR+/HER2-BC CAFs. (**c**) GSEA score from enriched KEGG pathways in each HR+/HER2-BC CAFs. (**d**) Trajectory analysis showed the pseudotime status of two types of iCAFs. (**e**) Heatmap of pseudotime-dependent gene expression in iCAF. (**f**) Kaplan–Meier survival analysis for TCGA-BRCA patients according to the deconvoluted fraction of each CAF. (**g**) Kaplan–Meier survival analysis for METABRIC breast cancer patients according to the deconvoluted fraction of each CAF.

**Figure 5 ijms-25-05876-f005:**
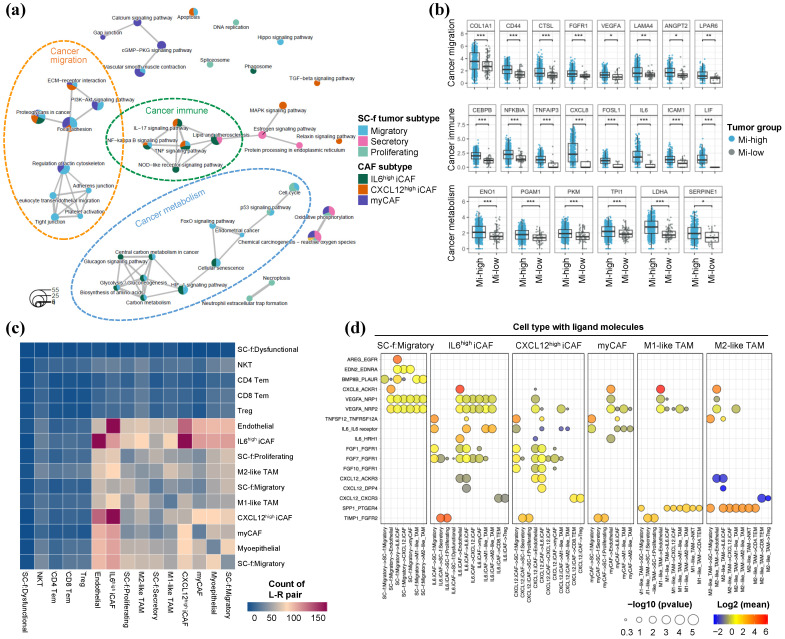
The diverse role of the migratory SC-f tumor subtype interacting with multiple CAFs. (**a**) Enrichment map displaying shared KEGG pathways based on DEGs from each SC-f tumor and CAF subtypes. Each node represents the enriched pathways, and a pie plot within the node indicates the proportion of member genes from each subtype. (**b**) Boxplots comparing the expression levels of functionally classified genes in fibroblast cells of scRNA-seq data between two groups: those with high migratory SC-f signature (Mi-high) and those with lower signature levels (Mi-low). Statistical significance was determined using the Wilcoxon signed-rank test. Statistical significance is indicated by asterisks: * *p* < 0.05, ** *p* < 0.01 and *** *p* < 0.001. (**c**) Cell–cell network according to ligand–receptor interaction predicted by CellphoneDB. The color indicates the number of matched L–R pairs between two cell types. (**d**) Dot plot showing the prevalence and expression of ligand and receptor interaction from tumor and TME cell types.

## Data Availability

Raw scRNA sequencing and processed data can be accessed from the NCBI Gene Expression Omnibus (GEO) database (accession code GSE 228499). The processed scRNA-seq used in Appendix A is available at GEO (accession code GSE113196). The bulk and SIDR-seq data used in this study are not publicly available due to confidentiality concerns about the patients’ medical information. However, interested researchers may request access to the data by contacting to K.Y.H., Samsung Genome Institute, Samsung Medical Center.

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
