# Peer review of "Migratory Tumor Cells Cooperate with Cancer Associated Fibroblasts in Hormone Receptor-Positive and HER2-Negative Breast Cancer"

_ijms, 2024, doi:10.3390/ijms25115876_

Round 1
Reviewer 1 Report
Comments and Suggestions for Authors
The manuscript exhibits potential and is captivating, showcasing various techniques employed by the authors. Nevertheless, there remains room for improvement.
I recommend acceptance pending revision, which I've outlined in the following points.
1) I would recommend selecting more specific markers to delineate populations, such as Luminal progenitor (KRT8/KRT19) and Luminal Differentiated (KRT4/PGR/PRLR/ESR1/CITED1), along with a more general marker like EPCAM for epithelial subsetting.
2) Is there co-expression of KRT5 with KRT14 in the population? Could it possibly represent a Basal population?
3) The definition of functional roles in Figure 2a, apart from dysfunctional and proliferating, lacks clarity.
4) Given that KRT18 is a Luminal marker, it might not be suitable for characterizing the migratory phenotype, which is typically associated with basal cells or luminals expressing KRT14.
5) In Figure S4a, the distinction between migratory and secretory phenotypes appears subtle, with the most notable contrast observed between proliferating cells and other phenotypes.
6) Figure 2k lacks clarity.
7) Figure 3b fails to demonstrate a discernible pattern of chromosome gain/loss.
8)Rows 154-160 lack clarity.
9) Dividing Figure 3c and S6b seems unnecessary since they contribute to the same result.
10)A suggestion may be to construct a 2-factor Kaplan-Meier analysis using TCGA and METABRIC data for Figure 3d.
11) Regarding classification, fibroblasts expressing CXCL12 could potentially be classified as ECM-CAFs in Figure 4a.
12) In Figure 5a, the only combination displaying specificity to a pathway appears to be migratory and IL6 in iCAFs.
13) The origin of markers in Figure S8b needs clarification.
14) The methodology for applying the PAM50 classification to cells requires elucidation. Was a pseudobulk constructed followed by a specific function, or was another pipeline utilized?
Overall, the paper's writing needs improvement, and figures should be aligned with the text to facilitate seamless comprehension for readers, avoiding the need to jump between figures.
Reviewer 2 Report
Comments and Suggestions for Authors
Dear Authors,
Here are my commentaries on your mansucript:
1. Introduction lacks paragraph related to objectives.
2. Rows 53-64 are more like a conclusion and probably would fit better in that section
3. The number of patients was extremely low considering that it is a common breast cancer subtype
4. The Results section precedes the MAterials and Methods section (this is wrong)
5. Please add trial limitations and, also add Conclusions section
The manuscript has serious editing flaws and needs consistent revision from the authors.
Round 2
Reviewer 1 Report
Comments and Suggestions for Authors
The revised manuscript address my previous concerns.
Author Response
Dear Reviewer 1,
Thank you for your prompt response and for acknowledging that the revised manuscript has addressed your previous concerns. Your feedback has been invaluable in refining our work, and we sincerely appreciate your evaluation of our manuscript.
We are committed to ensuring the quality and integrity of our research and welcome any additional feedback you may have.
Thank you once again for your time and consideration.
Warm regards,
Kyung Yeon Han
Reviewer 2 Report
Comments and Suggestions for Authors
Dear authors,
Thank you for your appreciation and effort regarding my suggestions.
However, your manuscript in its current form is still not ready for publication. It seems more like a presentation of intermediary data.
Please reconsider it after an increase in patient population.
